# Inter-rater reliability of a novel objective endpoint for benign central airway stenosis interventions: Segmentation-based volume rendering of computed tomography scans

**Ankush P. Ratwani**[1]*, **Heidi Chen**[2], **Leah Brown**[1], **Evan A. Schwartz**[3], **Khushbu Patel**[1], **Adam Guttentag**[4], **Thomas A. McLaren**[4], **Kim L. Sandler**[4], **Otis B. Rickman**[1], **Samira Shojaee**[1], **Robert J. Lentz**[1], **Fabien Maldonado**[1]

**1** Department of Medicine, Division of Allergy, Pulmonary and Critical Care, Vanderbilt University Medical Center, Nashville, TN, United States of America, **2** Department of Biostatistics, Vanderbilt University Medical Center, Nashville, TN, United States of America, **3** Department of Medicine, Division of Pulmonary, Allergy and Critical Care Medicine, Duke University School of Medicine, Durham, NC, United States of America, **4** Department of Radiology and Radiological Science, Vanderbilt University Medical Center, Nashville, TN, United States of America

* ankush.ratwani@vumc.org

**Data Availability Statement:** Data are available from Dryad. doi:10.5061/dryad.vmcvdncxq.

## Abstract

### Objectives

To evaluate the reliability of a novel segmentation-based volume rendering approach for quantification of benign central airway obstruction (BCAO).

### Design

A retrospective single-center cohort study.

### Setting

Data were ascertained using electronic health records at a tertiary academic medical center in the United States.

### Participants and inclusion

Patients with airway stenosis located within the trachea on two-dimensional (2D) computed tomography (CT) imaging and documentation of suspected benign etiology were included. Four readers with varying expertise in quantifying tracheal stenosis severity were selected to manually segment each CT using a volume rendering approach with the available free tools in the medical imaging viewing software OsiriX (Bernex, Switzerland). Three expert thoracic radiologists were recruited to quantify the same CTs using traditional subjective methods on a continuous and categorical scale.

**Funding:** This study was Supported by the Carol Odess Discovery Grant in Interventional Pulmonology.

**Competing interests:** The authors have declared that no competing interests exist.

## Outcome measures

The interrater reliability for continuous variables was calculated by the intraclass correlation coefficient (ICC) using a two-way mixed model with 95% confidence intervals (CI).

## Results

Thirty-eight patients met the inclusion criteria, and fifty CT scans were selected for measurement. The most common etiology of BCAO was iatrogenic in 22 patients (58%). There was an even distribution of chest and neck CT imaging within our cohort. The average ICC across all four readers for the volume rendering approach was 0.88 (95% CI, 0.84 to 0.93), suggesting good to excellent agreement. The average ICC for thoracic radiologists for subjective methods on the continuous scale was 0.38 (95% CI, 0.20 to 0.55), suggesting poor to fair agreement. The kappa for the categorical approach was 0.26, suggesting a slight to fair agreement amongst the raters.

## Conclusion

In this retrospective cohort study, agreement was good to excellent for raters with varying expertise in airway cross-sectional imaging using a novel segmentation-based volume rendering approach to quantify BCAO. This proposed measurement outperformed our expert thoracic radiologists using conventional subjective grading methods.

## Introduction

Benign central airway obstruction (BCAO) comprises a complex and multifactorial set of conditions [1]. Patients typically present with signs and symptoms of airflow limitation (dyspnea, cough, wheezing, stridor). However, given the relatively late onset of symptoms, up to half of patients present in respiratory distress [2]. The most common etiology is post-traumatic from prolonged endotracheal intubation or tracheostomy [3]. With the recent worldwide pandemic of severe acute respiratory syndrome coronavirus 2 (SARS-CoV-2), there have been reports of increasing BCAO cases following prolonged intubation [4, 5] with an expected increase in the coming years.

The burden of BCAO on patients and the healthcare system has been recently examined [6–9]. When examining a quality healthcare record, a study reported that patients with tracheal stenosis from prolonged intubation had an increased hospital stay (6.3 days; 95% CI, 6.0 to 6.3), in addition to an increase in hospital costs ($10,375; 95% CI, $9762 to $10,988). Another study reported that patients with post-intubation tracheal stenosis (PITS) that underwent nonsurgical treatments (Montgomery T-tube, silicone stent, or tracheostomy) had a decreased quality of life in the domains of physical limitation, bodily pain, and increased emotional distress following the procedure. Thus, early identification and shared decision-making regarding management are vital to prevent further patient morbidity.

However, during the last decade, comparative effectiveness studies evaluating novel therapeutic interventions for BCAO have been lacking, with management currently established on expert opinion and small retrospective cohort studies and case reports [10–12]. In addition, these studies have primarily used conventional subjective grading and classification systems as endpoints of disease recurrence, making interpretation of the results challenging due to

uncertain reliability. The field urgently needs more objective methods to assess airway luminal narrowing, which are reliable and not overly complex.

OsiriX (Berenex, Switzerland) is a Digital Imaging and Communications in Medicine (DICOM) viewer with the ability to perform advanced post-processing techniques on 2-dimensional (2D) computed tomography (CT) scans. In recent years, the ability of the software to create three-dimensional (3D) reconstructions of solid organs with volume rendering and segmentation-based techniques that are free of charge has generated excitement for preoperative planning and trainee simulation [13–15]. However, its use within the trachea has not been well described. Currently, it is being explored as an objective endpoint to quantify stenosis recurrence in an ongoing pilot randomized clinical trial (NCT04996173) evaluating the utility of adding spray cryotherapy to standard of care interventions in BCAO.

The objectives of our study are twofold. Firstly, we sought to evaluate the reliability of this novel objective manual segmentation-based volume rendering approach to quantify BCAO in raters with varying expertise in cross-sectional airway imaging. Secondly, we aimed to evaluate agreement amongst expert thoracic radiologists using conventional, subjective methods for assessing stenosis severity currently used in clinical practice and research.

## Methods

### Study subjects

We utilized an ongoing interventional pulmonary procedural database at Vanderbilt University Medical Center (VUMC) to identify patients for inclusion. Demographic data, including age, sex, smoking status, etiology of stenosis, CT imaging type (chest or neck), and axial slice thickness, were ascertained from the electronic health record (EHR). We included only patients with airway stenosis localized to the trachea and documentation of a suspected benign etiology within six months of the selected CT scan. The deidentified data were collected and managed using the Research Electronic Data Capture (REDCap) system [16, 17]. This study was approved by our local institutional review board (IRB #211567).

### CT image acquisition, segmentation, and volumetric analysis

All images were uploaded from the local picture archiving and communication system (PACS) to version 12.0 of OsiriX. After reviewing multiple sets of imaging before collection, a 3 cm measurement was felt to adequately capture the entire length of a stenotic segment in 95% of patients. The nadir stenosis point was identified in the soft tissue window and marked in the sagittal plane. We then measured 1.5 cm above and below this point. The airway lumen boundaries were then circumferentially marked using the closed polygon tool in the axial window. Four additional segments are manually segmented: the proximal end, one-third down the stenotic segment, two-thirds down the stenotic segment, and the most distal end (Fig 1). Finally, using the built-in repulsor function, the boundaries of the missing segments were manually adjusted to achieve a luminal fit. The resulting volumetric reconstruction is then generated and can be manipulated in 3D space with the resulting volume measurement (Fig 2).

### Interrater reliability

To assess the reliability of this novel endpoint, we identified four clinicians with different levels of expertise to interpret airway cross-sectional imaging and the ability of this approach to quantify airway stenosis severity. At the time of data collection, observer 1 (AR) was a pulmonologist, observers 2 (ES) and 3 (LB) were medicine residents, and observer 4 (KP) was a radiologist. AR gave each observer a thirty-minute introduction to the software with training on

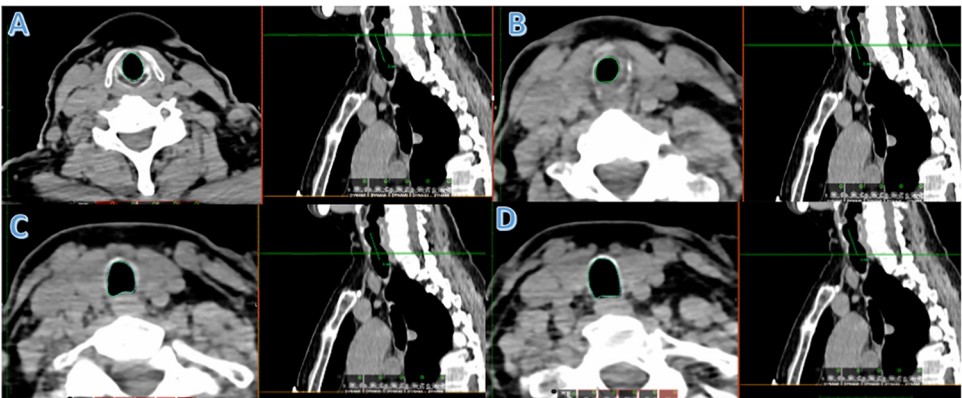

**Fig 1. The trachea is manually segmented for 3 cm along the superior-inferior axis in the axial view. (A)** Represents the most proximal segmentation. **(B)** Shows segmentation at the focal nadir point of stenosis. **(C)** Shows the two-thirds segmentation point. **(D)** Represents the most distal portion of the stenotic segment to be measured.

measurement and rendering of a test image. To provide for consistency of measurements AR marked the nadir point of stenosis on each image to be used as a reference. A screen recording was also available to all readers during the study to be used as a reference. The expectation was for all readers to start measuring within 24 hours of the training, and complete within a two-week timeframe.

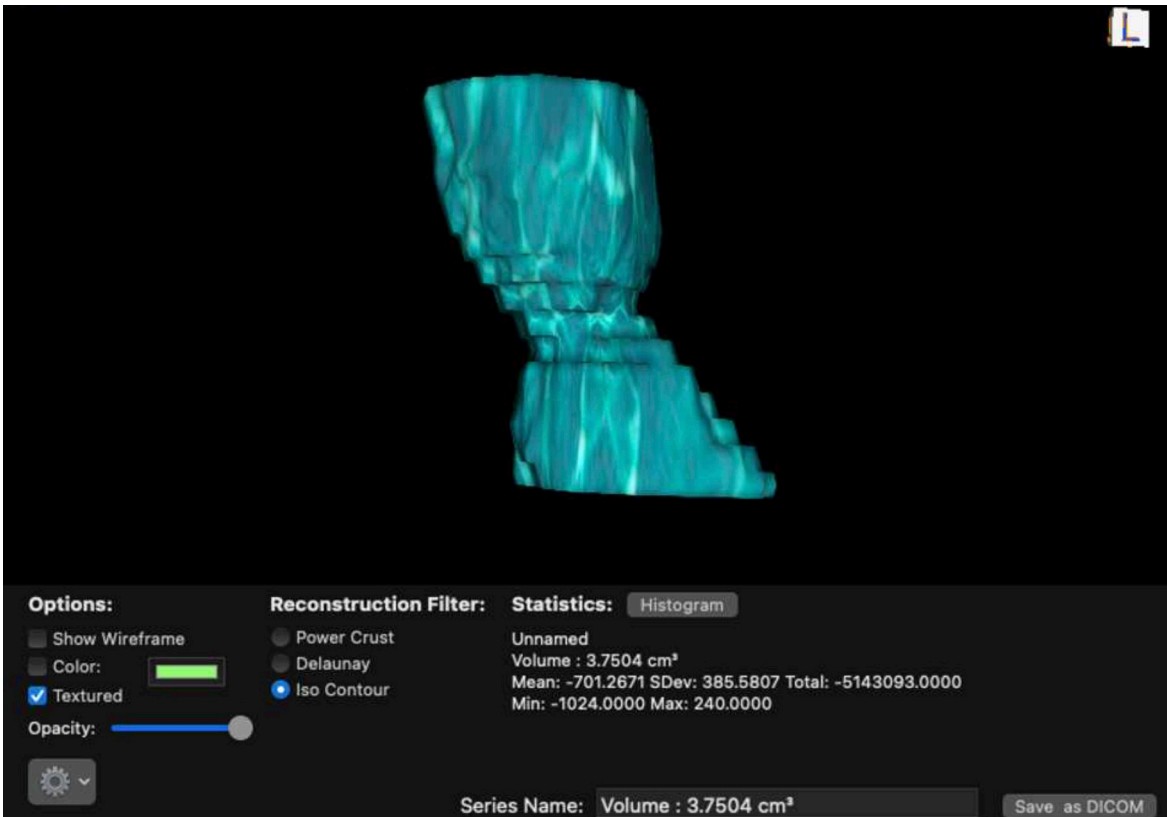

**Fig 2. The final 3D reconstruction of the trachea based on the measured regions of interest.** A resultant volume is displaced in cm³ with surrounding point statistics.

To compare the reproducibility of our segmentation-based approach with more subjective quantification methods, we recruited three expert thoracic radiologists (AG, KS, and TM) to read the same CT images and give their opinion of stenosis severity on both a continuous scale from 0–100% and a categorical scale using the Cotton-Myer grading system (grade 1: 0–50%, grade 2: 51–70%; grade 3: 71–99%; grade 4: No detectable lumen). In contrast to the objective methods, no nadir point of stenosis was pre-identified in the subjective approach.

## Statistical analysis

Descriptive statistics are presented, including means, medians, interquartile ranges (IQR), standard deviations, and ranges for continuous parameters and percentages and frequencies for categorical parameters. The interrater reliability for continuous variables was calculated by the intraclass correlation coefficient (ICC) using a two-way mixed model with 95% confidence intervals (CI). Fleiss's kappa was used to determine the reliability between groups of categorical variables. Guidelines for the interruption and reporting of ICC and Kappa have been previously described18,19. No correction was made for missing data. All analyses were performed by an independent statistician using R software, version 4.2.0 (R Foundation for Statistical Computing, www.r-project.org).

## Results

Thirty-eight patients met the inclusion criteria, with fifty CT scans between 2009 and 2021. Twenty-two (58%) were labeled iatrogenic (post-intubation or post-tracheostomy), 10 (26%) idiopathic, and the remaining six were due to other suspected etiologies (Table 1). Twenty-seven (71%) patients were women and most were never-smokers (68%). The resolution of the scans ranged from 1 to 5 mm, the median 2 mm (IQR, 1.25 to 3). Our cohort had an even distribution of CT neck and chest imaging. The calculated airway luminal volume means and standard deviations for each rater are displayed in Table 2. The average ICC across all four readers was 0.88 (95% CI, 0.84 to 0.93), suggesting good to excellent agreement. The average ICC for the thoracic radiologist on the continuous grading scale for the subjective approach was 0.38 (95% CI, 0.20 to 0.55), suggesting a poor to fair agreement. The average Fleiss kappa for the categorical Cotton-Myer grading system was 0.26, suggesting a slight to fair agreement amongst raters.

## Discussion

In this retrospective cohort study, we show that clinicians with varying expertise in airway cross-sectional imaging have good to excellent agreement when using a novel, objective segmentation-based volume approach to quantifying BCAO. Further, we demonstrate that when a group of expert thoracic radiologists evaluates the same images using traditional, more subjective approaches, they have poor overall agreement with both continuous and categorical measures.

Existing classification and grading systems [18–21] for BCAO are almost never used in clinical practice and only inconsistently in research. We believe reasons include 1) heavy reliance on a subjective interpretation of airway luminal narrowing, 2) poor external validation limiting generalizability, 3) lack of reproducibility, 4) poor standardization across and within specialties, 5) and poor correlation with physiological markers of disease activity and patient-reported outcomes (PRO). For example, changes in peak expiratory flow, which has been previously shown to predict disease recurrence in patients with BCAO, were recently shown to have a poor correlation with stenosis severity using the Cotton-Myer classification system with an overall kappa of 0.37 [22, 23].

**Table 1. Baseline characteristics with median (IQR) for continuous variables and number of patients with relative frequencies (%) for categorical variables.** †Iatrogenic includes post-intubation and post tracheostomy-induced stenosis. § Idiopathic includes no formal etiology given within a six-month time frame of the subject CT.

| Demographics | Total case data available (n) | |
|---|---|---|
| Age, years | 38 | 48 (35–60) |
| Gender | 38 | |
| Female | | 29 (76%) |
| Smoking Status, | 38 | |
| Former | | 12 (32%) |
| CT Type | 50 | |
| CT Chest | | 25 (50%) |
| CT Neck | | 25 (50%) |
| Axial Slice Thickness (mm) | 50 | |
| 2 | | 15 (30%) |
| 1 | | 10 (20%) |
| 3 | | 7 (14%) |
| 2.5 | | 6 (12%) |
| 1.25 | | 5 (10%) |
| 5 | | 2 (4%) |
| 3.75 | | 4 (8%) |
| 1.5 | | 1 (2%) |
| Etiology of BCAO | 38 | |
| Iatrogenic† | | 22 (58%) |
| Idiopathic§ | | 10 (26%) |
| Radiation | | 2 (5%) |
| Reflux | | 2 (5%) |
| Vasculitis | | 1 (3%) |
| Fibrosing Mediastinitis | | 1 (3%) |

For several reasons, classification and grading systems for BCAO that rely heavily on subjective evaluation measures prove to be the most problematic. *First*, they may not correctly characterize complex lesions, such as lesions with a more significant vertical extent ($\geq$ 1 cm in length), those that invade surrounding cartilaginous structures, and those with dynamic collapse from underlying malacia. *Second*, reproducibility amongst providers is challenging as assessment of luminal narrowing is often "eye-balled," with difficulties with interpreting when a transition state occurs (e.g., 49% versus 50% stenosis). *Finally*, most systems that use a visual grading of stenosis require direct visualization with inherent procedural and anesthesia risks.

We believe that an approach using volumetric assessment to quantify airway stenosis from readily available 2D CT imaging may address some of these challenges. By visualizing a stenotic segment in multiple dimensions, one can get a more comprehensive sense of the lesion's vertical and structural extent. This can prove advantageous in decision-making regarding early

**Table 2. Mean and standard deviation (SD) for generated tracheal volumes for each rater overall CTs.**

| Rater | Mean (cm³) | SD (cm³) |
|---|---|---|
| AR | 3.41 | 1.21 |
| ES | 3.67 | 1.36 |
| LB | 3.80 | 1.26 |
| KP | 3.34 | 1.31 |

referral for surgical resection, as this has shown to be the most definitive treatment in patients with BCAO [24] who are suitable candidates. Additionally, objective data following an endoscopic therapeutic intervention allows for better identification of disease recurrence and need for a repeat procedure. Finally, little expertise in measurement is required as we have shown that readers with minimal training and expertise in airway cross-sectional imaging were able to have strong agreement. This contrasts with our radiologists' subjective grading, highlighting the challenges with current approaches in everyday practice.

This study has several notable strengths. Multiple etiologies of BCAO were included, highlighting the generalizability of these findings. Our reported ICC suggests that this novel measurement is reliable regardless of the underlying type of CT performed (chest or neck). The patients in our cohort had a variety of stenotic lengths and severity of luminal narrowing, highlighting the ability of our readers to agree with lesions of different complexities. The ability to analyze these images using the free of charge tools in OsiriX with relatively brief training suggests that this approach could be widely adopted with minimal cost or effort. However, it is essential to consider the inherent trade-off of increased time required to perform such measurements compared to traditional subjective methods.

Limitations of this study include a modest sample size and testing limited to the confines of the trachea. The a priori identification of the point of nadir stenosis may have introduced bias, improving recognition of the stenotic area. However, this approach was chosen to minimize ambiguity in identifying structural abnormalities and prioritize accurate measurements. As the dataset was retrospective, direct correlation of tracheal volume with stenosis severity or quality of life measures was not possible. Future research could explore establishing thresholds or criteria for tracheal volume indicative of stenosis severity or its impact on quality of life. Additionally, the study was limited in assessing dynamic imaging or individuals with underlying malacia.

In conclusion, we report the reliability of a novel objective measure for quantifying airway stenosis based on a straightforward volume rendering approach in the open-source medical imaging viewer OsiriX. This measure holds promise as an objective research endpoint for assessing airway luminal narrowing and may serve as an accurate assessment of disease recurrence in studies testing new therapeutic interventions in BCAO.

## Author Contributions

**Conceptualization:** Ankush P. Ratwani, Samira Shojaee, Robert J. Lentz, Fabien Maldonado.

**Data curation:** Ankush P. Ratwani, Leah Brown, Evan A. Schwartz, Khushbu Patel, Adam Guttentag, Thomas A. McLaren, Kim L. Sandler.

**Formal analysis:** Ankush P. Ratwani, Heidi Chen.

**Investigation:** Ankush P. Ratwani, Fabien Maldonado.

**Methodology:** Heidi Chen, Robert J. Lentz, Fabien Maldonado.

**Supervision:** Fabien Maldonado.

**Visualization:** Fabien Maldonado.

**Writing – original draft:** Ankush P. Ratwani.

**Writing – review & editing:** Ankush P. Ratwani, Leah Brown, Evan A. Schwartz, Khushbu Patel, Adam Guttentag, Thomas A. McLaren, Kim L. Sandler, Otis B. Rickman, Samira Shojaee, Robert J. Lentz, Fabien Maldonado.

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
