## [Decision Letter · Decision Letter 0]

6 Jul 2023

PONE-D-23-03574Inter-rater reliability of a novel objective endpoint for benign central airway stenosis interventions: segmentation-based volume rendering of computed tomography scansPLOS ONE

Dear Dr. Ratwani,

Thank you for submitting your manuscript to PLOS ONE. After careful consideration, we feel that it has merit but does not fully meet PLOS ONE’s publication criteria as it currently stands. Therefore, we invite you to submit a revised version of the manuscript that addresses the points raised during the review process.

We look forward to receiving your revised manuscript.

Kind regards,

Yih Miin Liew

Academic Editor

PLOS ONE

Journal Requirements:

Reviewers' comments:

Reviewer's Responses to Questions

**Comments to the Author**

1. Is the manuscript technically sound, and do the data support the conclusions?

Reviewer #1: Yes

Reviewer #2: Partly

2. Has the statistical analysis been performed appropriately and rigorously? 

Reviewer #1: Yes

Reviewer #2: Yes

3. Have the authors made all data underlying the findings in their manuscript fully available?

Reviewer #1: Yes

Reviewer #2: No

4. Is the manuscript presented in an intelligible fashion and written in standard English?

Reviewer #1: Yes

Reviewer #2: Yes

5. Review Comments to the Author

Reviewer #1: Many thanks for asking me to review this manuscript.

This is a great study. My only reservations are that the study would have benefited from a larger cohort of patients, with a wider audience (perhaps international) to quantify the variables.

I suspect this study is a pilot study of some sorts - leading to a wider and bigger study.

Reviewer #2: This paper assesses the inter-rater reliability of a new method of BCAO quantification, comparing it with two conventional visual methods of assessment.

Table 1: indicate which variables are continuous. Also, some percentages appear to be wrongly calculated (for axial slice thickness).

Methods: the method requires identifying the nadir stenosis point. Was each rater required to locate this point themselves, or was it specified for them?

Methods: the method involves measuring +/- 1.5cm above and below the nadir stenosis point. However, some of the volumes appear to have slice thickness >3 mm. How were they handled?

Methods: the new method measures absolute volume, but the conventional methods are based on relative cross-sectional area (stenosis severity). I do believe a segmentation method would be more accurate, but a specific formula quantifying stenosis severity from the segmentations should be developed and tested.

Methods and results: the new segmentation method would likely take longer to perform than the conventional visual methods. The time taken for assessment should be measured to understand the trade-off.

General: OsiriX is called "open source" in multiple locations in the text. OsiriX may have started off as open source, but its current incarnation is commercial software.

6. PLOS authors have the option to publish the peer review history of their article (what does this mean?). If published, this will include your full peer review and any attached files.

Reviewer #1: No

Reviewer #2: No

---

## [Author Response · Author response to Decision Letter 0]

12 Jul 2023

Response to reviewers:

Reviewer #1: Many thanks for asking me to review this manuscript.

This is a great study. My only reservations are that the study would have benefited from a larger cohort of patients, with a wider audience (perhaps international) to quantify the variables.

I suspect this study is a pilot study of some sorts - leading to a wider and bigger study.

Response: We sincerely appreciate the reviewer's careful evaluation of our manuscript. We acknowledge the valid concern raised regarding the small size of our study population. It is important to note that benign airway stenosis is an uncommon disorder, with even major academic centers encountering fewer than 50 patients per year with this diagnosis. In light of this rarity, our data represents a comprehensive summary of patients spanning a 10-year period, thereby highlighting the scarcity of available cases and CT imaging. Furthermore, we would like to emphasize that this study serves as a pilot investigation with two primary objectives. Firstly, it serves as the primary outcome measure it serves as the primary outcome measure for a pilot multicenter randomized controlled trial aimed generate preliminary data on a particular treatment for subglottic stenosis and validate this endpoint. Secondly, it is part of a larger ongoing effort at our institution to prospectively collect data on future patients, with the intention of collaborating with national and international centers in the future.

Reviewer #2: This paper assesses the inter-rater reliability of a new method of BCAO quantification, comparing it with two conventional visual methods of assessment.

Table 1: indicate which variables are continuous. Also, some percentages appear to be wrongly calculated (for axial slice thickness).

Response: We thank the reviewer for bringing this to our attention, we have corrected the percentages and have made it more clear which variables are continuous (age). 

Methods: the method requires identifying the nadir stenosis point. Was each rater required to locate this point themselves, or was it specified for them?

Response: To provide for consistency of measurements between readers, the reader AR marked the nadir point of stenosis on each image to be used as a reference. We have added this explanation in our methods. 

Methods: the method involves measuring +/- 1.5cm above and below the nadir stenosis point. However, some of the volumes appear to have slice thickness >3 mm. How were they handled?

Response: We apologize for any confusion caused by our previous description. To clarify, the measurement of +/- 1.5cm above and below the nadir stenosis point does not directly pertain to CT slice thickness but rather refers to the spatial extent within the airway that was assessed. CT slice thickness is a distinct technical parameter that determines the thickness of the individual image slices acquired during the CT scan, as mentioned previously. It does not directly impact the measurement technique employed in our study. This measurement was performed regardless of the specific CT slice thickness utilized during the scanning process. The purpose was to assess the anatomical characteristics and dimensions of the stenosis within a defined region, irrespective of the thickness of individual slices in the CT dataset.

Methods: the new method measures absolute volume, but the conventional methods are based on relative cross-sectional area (stenosis severity). I do believe a segmentation method would be more accurate, but a specific formula quantifying stenosis severity from the segmentations should be developed and tested.

Response: We appreciate the reviewer's valuable comments. We acknowledge the limitations associated with traditional methods, which often pose challenges in terms of accuracy, reproducibility, and ease of use. In light of these considerations, we opted for the segmentation method employed in our study. This approach capitalizes on the utilization of readily available CT images within the clinical workflow, offering the advantage of being easily performed by readers with varying levels of expertise in interpreting airway cross-sectional images. A formula-based approach could possibly be evaluated in future studies. 

Methods and results: the new segmentation method would likely take longer to perform than the conventional visual methods. The time taken for assessment should be measured to understand the trade-off.

Response: Although the segmentation method may require slightly more time for implementation, its ease of use and straightforward measurement steps have resulted in relatively high agreement among users. Considering this positive outcome, it would be intriguing to explore the aspect of time in future studies. However, we would like to reemphasize the limitations associated with conventional visual methods. These drawbacks encompass poor relative accuracy, limited reproducibility, difficulties in comprehending the stenotic structure in three-dimensional space, challenges in effectively communicating measurements, and limitations in tracking temporal changes over time. By highlighting these issues, we underscore the need for alternative approaches such as the segmentation method employed in our study to overcome these shortcomings and enhance the reliability and comprehensiveness of airway stenosis assessments.

General: OsiriX is called "open source" in multiple locations in the text. OsiriX may have started off as open source, but its current incarnation is commercial software.

Response: We appreciate the reviewer's valid observation. However, we would like to clarify that all the measurement components and aspects of our study were conducted using the open-source functionality of OsiriX. No readers involved in our study incurred any expenses for commercial software usage. We ensured that our measurements were performed solely with the freely available tools and features of OsiriX, thereby promoting accessibility and eliminating any potential financial barriers for the readers involved in our research.

---

## [Decision Letter · Decision Letter 1]

26 Jul 2023

PONE-D-23-03574R1Inter-rater reliability of a novel objective endpoint for benign central airway stenosis interventions: segmentation-based volume rendering of computed tomography scansPLOS ONE

Dear Dr. Ratwani,

Thank you for submitting your manuscript to PLOS ONE. After careful consideration, we feel that it has merit but does not fully meet PLOS ONE’s publication criteria as it currently stands. Therefore, we invite you to submit a revised version of the manuscript that addresses the points raised during the review process.

We look forward to receiving your revised manuscript.

Kind regards,

Jorge Spratley, MD, PhD

Academic Editor

PLOS ONE

Journal Requirements:

Additional Editor Comments:

Please address all the reviewers' comments in a concise fashion. As stated by Reviewer #2, some limitations of the study should be clearly stated in the body of the manuscript. The interesting remark by Reviewer #3, concerning the comparison of imaging measurements with the gold-standard endoscopic direct measurement of the stenosis, should also be commented.

Reviewers' comments:

Reviewer's Responses to Questions

**Comments to the Author**

1. If the authors have adequately addressed your comments raised in a previous round of review and you feel that this manuscript is now acceptable for publication, you may indicate that here to bypass the “Comments to the Author” section, enter your conflict of interest statement in the “Confidential to Editor” section, and submit your "Accept" recommendation.

Reviewer #1: All comments have been addressed

Reviewer #2: (No Response)

Reviewer #3: (No Response)

2. Is the manuscript technically sound, and do the data support the conclusions?

Reviewer #1: Yes

Reviewer #2: Partly

Reviewer #3: Yes

3. Has the statistical analysis been performed appropriately and rigorously? 

Reviewer #1: Yes

Reviewer #2: Yes

Reviewer #3: Yes

4. Have the authors made all data underlying the findings in their manuscript fully available?

Reviewer #1: Yes

Reviewer #2: No

Reviewer #3: Yes

5. Is the manuscript presented in an intelligible fashion and written in standard English?

Reviewer #1: Yes

Reviewer #2: Yes

Reviewer #3: Yes

6. Review Comments to the Author

Reviewer #1: The author addressed my concerns and provided clear explanations and reasoning in the revised manuscript.

Reviewer #2: =====

> To provide for consistency of measurements between readers, the reader AR marked the nadir point of stenosis on each image to be used as a reference. We have added this explanation in our methods.

I understand why this was done, but unfortunately this limits the validity of the interrater measurement, particularly when the paper claims that “readers with minimal training and expertise in airway cross-sectional imaging were able to have strong agreement”. At the least, this should be mentioned in the study limitations.

Also, was a similar reference location specified for the subjective method? Please state it similarly in the text.

=====

> To clarify, the measurement of +/- 1.5cm above and below the nadir stenosis point does not directly pertain to CT slice thickness but rather refers to the spatial extent within the airway that was assessed.

My apologies, I somehow misread the original text as +/- 1.5mm (not cm) above and below the nadir stenosis point.

=====

> A formula-based approach could possibly be evaluated in future studies.

The point I'm making is that both methods are measuring different outcomes. The subjective method explicitly assesses stenotic severity (normal vs. stenotic), whereas the objective method only measures total volume, with no assessment of stenosis. E.g., if a particular volume threshold is used to diagnose stenosis, there's no indication how the interrater variability would affect such a diagnosis. At the least, this should be mentioned in the study limitations.

=====

> Although the segmentation method may require slightly more time for implementation, its ease of use and straightforward measurement steps have resulted in relatively high agreement among users.

I agree that the higher accuracy would likely be valuable. However, the objective measurements do take more time and the trade-off should be made clear. Anyway, the lack of time measurements should be mentioned in the study limitations.

=====

> we would like to clarify that all the measurement components and aspects of our study were conducted using the open-source functionality of OsiriX.

“Open source” carries a specific meaning, that the source code is made available. As far as I’m aware, the last open source version of OsiriX was v5.8 (I think the current version is v12). You should probably use the term “free of charge” instead.

Reviewer #3: The manuscript is interesting, addressing a new diagnosis method for airway stenosis. It would benefit of a larger cohort to stregthen the data, however I recognize that airway stenosis is a rare pathology.

You only reported the imaging evaluation. It would be of great value to compare these results with the endoscopy findings. Do you have these data?

In the last paragraph of the section "Statistical analysis", references 18 and 19 should be above the line, as the remaining references of the manuscript. The hyperlink of the website mentioned should also be removed.

7. PLOS authors have the option to publish the peer review history of their article (what does this mean?). If published, this will include your full peer review and any attached files.

Reviewer #1: No

Reviewer #2: No

Reviewer #3: No

---

## [Author Response · Author response to Decision Letter 1]

5 Aug 2023

Response to reviewers:

Reviewer #2: 

1. To provide for consistency of measurements between readers, the reader AR marked the nadir point of stenosis on each image to be used as a reference. We have added this explanation in our methods.

I understand why this was done, but unfortunately this limits the validity of the interrater measurement, particularly when the paper claims that “readers with minimal training and expertise in airway cross-sectional imaging were able to have strong agreement”. At the least, this should be mentioned in the study limitations.

Also, was a similar reference location specified for the subjective method? Please state it similarly in the text.

Response: We express our gratitude to the reviewer for their valuable comments. The remark regarding the measurement of the nadair point as a reference is well-taken, and we acknowledge that it may somewhat restrict the validity of the interrater measurements. Nevertheless, this approach was adopted to minimize ambiguity when identifying structural abnormalities, with a primary focus on accurate measurements. However, it is worth noting that this approach was not employed for the subjective method. We will duly acknowledge this limitation in our study.

2. A formula-based approach could possibly be evaluated in future studies. 

The point I'm making is that both methods are measuring different outcomes. The subjective method explicitly assesses stenotic severity (normal vs. stenotic), whereas the objective method only measures total volume, with no assessment of stenosis. E.g., if a particular volume threshold is used to diagnose stenosis, there's no indication how the interrater variability would affect such a diagnosis. At the least, this should be mentioned in the study limitations.

Response: You are correct in pointing out that both methods in our study are measuring different outcomes. We acknowledge that this issue was not explicitly addressed in the study, and the topic of future work. Your suggestion to include this point in the study limitations is well-founded, and we agree that it is essential to highlight this potential limitation to provide a comprehensive understanding of our findings.

3. Although the segmentation method may require slightly more time for implementation, its ease of use and straightforward measurement steps have resulted in relatively high agreement among users.

I agree that the higher accuracy would likely be valuable. However, the objective measurements do take more time and the trade-off should be made clear. Anyway, the lack of time measurements should be mentioned in the study limitations.

Response: You raise an important point regarding the trade-off between higher accuracy and the additional time required for objective measurements. We completely agree that this trade-off should be transparently presented in our study to provide a comprehensive understanding of the implications of using objective measurements.

4. We would like to clarify that all the measurement components and aspects of our study were conducted using the open-source functionality of OsiriX.

“Open source” carries a specific meaning, that the source code is made available. As far as I’m aware, the last open source version of OsiriX was v5.8 (I think the current version is v12). You should probably use the term “free of charge” instead.

Response: Valid points, we will clarify and include the term that the version used was “free of charge”. 

Reviewer #3: 

1. You only reported the imaging evaluation. It would be of great value to compare these results with the endoscopy findings. Do you have these data?

Response: Thank you for raising a valuable point regarding the comparison of imaging evaluation with endoscopy findings. We understand the significance of such a comparison, as it can provide comprehensive insights into the clinical correlations and validate the accuracy of our imaging assessments. We do not have endoscopic images for most cases in our dataset. This is a common scenario in our study setting, as patients often undergo imaging without undergoing endoscopic intervention (surveillance). As a result, we were unable to directly compare the imaging results with endoscopy findings. This presents an opportunity for future research and correlation of tracheal volume to severity of stenosis on endoscopy. We will note this in our limitations. 

2. In the last paragraph of the section "Statistical analysis", references 18 and 19 should be above the line, as the remaining references of the manuscript. The hyperlink of the website mentioned should also be removed.

Response: Thank you, we will address this.

---

## [Editor Report · Decision Letter 2]

8 Aug 2023

Inter-rater reliability of a novel objective endpoint for benign central airway stenosis interventions: segmentation-based volume rendering of computed tomography scans

PONE-D-23-03574R2

Dear Dr. Ratwani,

We’re pleased to inform you that your manuscript has been judged scientifically suitable for publication and will be formally accepted for publication once it meets all outstanding technical requirements.

Kind regards,

Jorge Spratley, MD, PhD

Academic Editor

PLOS ONE

---

## [Editor Report · Acceptance letter]

16 Oct 2023

PONE-D-23-03574R2 

Inter-rater reliability of a novel objective endpoint for benign central airway stenosis interventions: segmentation-based volume rendering of computed tomography scans 

Dear Dr. Ratwani:

I'm pleased to inform you that your manuscript has been deemed suitable for publication in PLOS ONE. Congratulations! Your manuscript is now with our production department. 

Kind regards, 

on behalf of

Professor Jorge Spratley 

Academic Editor

PLOS ONE